# The association between number and ages of children and the physical activity of mothers: Cross-sectional analyses from the Southampton Women's Survey

Rachel F. Simpson[1]*, Kathryn R. Hesketh[1,2], Sarah R. Crozier[3,4], Janis Baird[3,5], Cyrus Cooper[3,5], Keith M. Godfrey[3,5], Nicholas C. Harvey[3,5], Kate Westgate[1], Hazel M. Inskip[3,5], Esther M. F. van Sluijs[1]

1 MRC Epidemiology Unit, University of Cambridge School of Clinical Medicine, Cambridge, United Kingdom, 2 UCL Great Ormond Street Institute of Child Health, London, United Kingdom, 3 MRC Lifecourse Epidemiology Centre (University of Southampton), Southampton General Hospital, Southampton, United Kingdom, 4 NIHR Applied Research Collaboration Wessex, Southampton, United Kingdom, 5 NIHR Southampton Biomedical Research Centre, University of Southampton and University Hospital Southampton NHS Foundation Trust, Southampton, United Kingdom

* rs2056@cam.ac.uk

**Data Availability Statement:** Due to ethical concerns, including promises made to participants at the outset of the study that the data would not be

## Abstract

### Background

Physical activity (PA) has many health benefits, but motherhood is often associated with reduced PA. Considering that ages and number of children may be associated with maternal PA, and that PA patterns may change as children transition to formal schooling, we aimed to investigate the associations between ages and number of children and device-measured maternal PA.

### Methods

Cross-sectional analyses were conducted using data from 848 mothers from the Southampton Women's Survey at two different timepoints. Two-level random intercept linear models were used to investigate associations between ages (≤4y(ears) ("younger"), school-aged, both age groups) and number (1, 2, ≥3) of children, and their interaction, and accelerometer-assessed minutes of maternal moderate or vigorous PA (log-transformed MVPA) and light, moderate or vigorous PA (LMVPA).

### Results

Women with any school-aged children engaged in more MVPA than those with only ≤4y (e.g. % difference in minutes of MVPA [95% confidence interval]: 46.9% [22.0;77.0] for mothers with only school-aged vs only ≤4y). Mothers with multiple children did less MVPA than those with 1 child (e.g. 12.5% [-1.1;24.3] less MVPA for those with 2 children). For mothers with multiple children, those with any school-aged children did less LMVPA than those with only ≤4y (e.g. amongst mothers with 2 children, those with only school-aged

made openly available, we are unable to share the data freely. The SWS team can provide the data on request subject to appropriate approvals. Researchers wishing to use the data would need to make a formal application to the MRC LEC Cohort Oversight Committee (mrclec@mrc.soton.ac.uk) and ensure appropriate ethical approval is in place. Subject to approval and formal agreements being signed, the data would then be provided.

**Funding:** RFS is funded by an ESRC studentship (RG84395). The work of EvS, KRH and KW is supported by the Medical Research Council (grant numbers MC_UU_00006/5 and MC_UU_12015/3). This work was undertaken under the auspices of the Centre for Diet and Activity Research (CEDAR) (grant number MR/K023187/1), a UKCRC Public Health Research Centre of Excellence. Funding from the British Heart Foundation, Cancer Research UK, Economic and Social Research Council, Medical Research Council, the National Institute for Health Research, and the Wellcome Trust, under the auspices of the UK Clinical Research Collaboration, is gratefully acknowledged. KRH is also funded by the Wellcome Trust (grant number 107337/Z/15/Z), and KW through the NIHR Cambridge Biomedical Research Centre (IS-BRC-1215-20014). KMG is supported by the National Institute for Health Research (NIHR Senior Investigator (NF-SI-0515-10042), NIHR Southampton 1000DaysPlus Global Nutrition Research Group (17/63/154) and NIHR Southampton Biomedical Research Centre (IS-BRC-1215-20004)) and the European Union (Erasmus+ Programme ImpENSA 598488-EPP-1-2018-1-DE-EPPKA2-CBHE-JP). The work of CC, KMG, NCH, HMI, JB and SRC was supported by funding from the Medical Research Council, British Heart Foundation, the UK Food Standards Agency, British Lung Foundation, the Arthritis Research UK, National Osteoporosis Society, International Osteoporosis Foundation, Cohen Trust, the European Union Seventh Framework Programme (FP7/2007-2013) Early Nutrition project under grant agreement 289346, and European Union's Horizon 2020 research and innovation programme under grant agreement No 733206, 9.6 M€ (LifeCycle), NIHR Southampton Biomedical Research Centre, and National Institute of Health Research Musculoskeletal Biomedical Research Unit, Oxford. The funders had no role in study design, data collection and analysis, decision to publish, or preparation of the manuscript.

**Competing interests:** I have read the journal's policy and the authors of this manuscript have the following competing interests: CC received personal fees from Alliance for Better Bone Health,

children did 34.0 [3.9;64.1] mins/day less LMVPA). For mothers with any ≤4y, those with more children did more LMVPA (e.g. amongst mothers with only ≤4y, those with 2 children did 42.6 [16.4;68.8] mins/day more LMVPA than those with 1 child).

## Conclusions

Mothers with multiple children and only children aged ≤4y did less MVPA. Considering that many of these women also did more LMVPA than mothers with fewer or older children, interventions and policies are needed to increase their opportunities for higher intensity PA to maximise health benefits.

## Trial registration

ClinicalTrials.gov Identifier: NCT04715945.

## Introduction

Physical activity has many health benefits [1]. It decreases the risk of multiple physical health outcomes, ranging from various cancers to cardiovascular disease [2]. It is also associated with weight maintenance [2], better mental health [2], and mitigation of negative effects of sedentary behaviour [3]. There are additional potential benefits of physical activity for parents. These include increased confidence in ability to cope with the daily challenges of being a parent, and strengthening of parent-child relationships through co-participation [4, 5]. There is also evidence that parental physical activity is positively associated with that of children, especially in studies using device-based measurements [6, 7]. However, despite the many potential gains which physical activity could bring to parents, they are less active than non-parents [8].

It is important to decipher what factors are associated with parental physical activity to understand this behaviour, identify sub-groups of parents who are more at risk of insufficient physical activity and determine which factors could be targeted to increase physical activity. Previous review-level evidence concerning factors, including ages and number of children, associated with parental or specifically maternal physical activity has mostly been inconclusive [8, 9]. Considering that ages and number of children are correlates specific to parents, rather than adults in general, and that previous reviews were inconclusive, it is important to further investigate the association between these factors and parental physical activity. The few studies that have used device-based physical activity assessment to investigate either the association between ages or number of children and maternal physical activity suggest that mothers of younger children do less moderate or vigorous physical activity (MVPA) than mothers of older children, but report mixed findings for the number of children [10–12].

Data from the UK Southampton Women's Survey (SWS) provide accelerometer-assessed physical activity in mothers, along with details of number of younger and older siblings of the index child at 4 years of age (4y) and 6 years of age (6y). Previous analyses in this cohort, when the index child was 4y, found that mothers who had younger children, as well as the index child, engaged in more light physical activity (LPA) [13], but no association was found with MVPA [13]. This work extends the previous analysis by examining the separate effects of number and ages of children, also determining whether there is an interaction between the two variables in relation to maternal physical activity. This allows us to identify which subgroups of mothers are more at risk of insufficient physical activity than others. In the UK, children start

Amgen, Eli Lilly, GSK, Medtronic, Novartis, Pfizer, Roche, Servier, Takeda and UCB. KMG has received reimbursement for speaking at conferences sponsored by companies selling nutritional products, and is part of an academic consortium that has received research funding from Abbott Nutrition, Nestec, BenevolentAI Bio Ltd. and Danone. NCH has received consultancy, lecture fees and honoraria from Alliance for Better Bone Health, AMGEN, MSD, Eli Lilly, Servier, UCS, Shire, Consilient Healthcare, Kyowa Kirin and Internis Pharma. The remaining authors declare they have no competing interests. This does not alter our adherence to PLOS ONE policies on sharing data and materials.

**Abbreviations:** 4y, 4 years old; 5y, 5 years old; 6y, 6 years old; 95%CI, 95% confidence interval; BMI, body mass index; CSE, certificate of secondary education; DAG, direct acyclic graph; GMR, geometric mean ratio; HND, higher national diploma; IQR, interquartile range; LMVPA, light, moderate or vigorous physical activity; LPA, light physical activity; MVPA, moderate or vigorous physical activity; SD, standard deviation; SWS, Southampton Women's Survey.

school in the September before they turn 5y [14]. It is plausible that maternal physical activity and opportunities to be active may change at this point as previous evidence suggests that the relationship between mother-child physical activity also changes during this transition [15]. It is therefore useful to compare the physical activity levels of mothers with school-aged vs younger children. Using data from the SWS, the aim of this cross-sectional study is to investigate the association between number and ages of children, categorised as ≥5y (hereafter referred to as school-aged) and ≤4y (hereafter referred to as younger children), and maternal accelerometer-assessed physical activity.

## Methods

These analyses are reported following the STROBE guidelines for cross-sectional studies (S1 Appendix) [16].

### Population

The SWS is a cohort study based in Southampton, UK. Between 1998 and 2002, non-pregnant women aged 20–34 years were recruited through general practices and participated in interviews [17]. Women who subsequently became pregnant were invited to take part, and live births were followed up at various intervals [13, 17]. Twins and higher-order pregnancies were not included in the study. A sub-study of mothers and the index child was conducted for children reaching 4y (mean age 4.1 (standard deviation (SD) 0.1)) between March 2006 and June 2009 to investigate their physical activity [15]. Mothers with index children born after January 2000 were then approached for a visit when the index child was 6–7 years (mean age 6.7 (SD 0.3) and hereafter referred to as 6y) between March 2007 and August 2012 [15]. The Southampton and South West Hampshire Local Research Ethics committee granted ethical approval for the study. Direct written consent was obtained from all women and written parental consent with verbal child assent was gained for children.

### Measures used in the current analyses

**Physical activity assessment and variables.** Maternal physical activity was assessed using a combined heart rate and uniaxial accelerometer (Actiheart, CamNtech, Cambridgeshire, UK) secured to the chest and set to record at 60-second epochs [15]. Participants were asked to wear the monitor continuously for seven days, including during water-based activity and sleep, and return them via secure post [15]. Only accelerometer data were used for these analyses due to the focus on physical activity intensity, and that heart rate data were not dynamically individually calibrated.

Data periods of ≥100 minutes with zero-activity were removed and recordings between 11pm and 6am were excluded to remove time spent sleeping [13]. Mothers were eligible for inclusion in analyses if they had ≥10 hours of valid accelerometer data for at least one day [13, 18]. Valid accelerometer data were available for 621 and 608 women at the 4y and 6y waves respectively (total n = 1034; n = 195 women with both timepoints).

Daily maternal MVPA and light, moderate or vigorous physical activity (LMVPA) were the outcome measures.

The main outcomes examined were average minutes of MVPA and LMVPA per day, on weekdays and on weekend days. Exploratory outcome variables were average daily minutes MVPA and LMVPA for periods when mothers might be more or less likely to be with their children: 6-9am (weekday morning); 9am-3pm (weekday school/work day); 3-7pm (weekday late afternoon); 7-11pm (weekday evening) and weekends: 6am-7pm (weekend day); 7-11pm (weekend evening). MVPA was chosen as an outcome as it has been shown to have additional

health benefits to LPA even when volume of physical activity energy expenditure is taken into consideration [19]. However, LMVPA was also chosen as an outcome as a measure of overall physical activity since it may be more likely that mothers of young children engage in greater LPA than MVPA, and LPA has also been shown to have health benefits [20]. Cut-points for MVPA ($\geq$400cpm) and LMVPA ($\geq$20cpm) were applied having been scaled using a conversion factor of 5 from the Actiheart accelerometer as previously validated [18, 21]. The Actiheart intensity thresholds equate to 2000 counts for MVPA and 100 counts for LPA in the Actigraph 7164 accelerometer (Actigraph, Pensacola, FL) [21, 22].

Other physical activity variables derived for descriptive purposes were: meeting physical activity guidelines ($\geq$150 minutes of MVPA/week, or a mean of $\geq$21.4 mins/day), valid days of collection, and average daily wear-time.

**Exposure variables.** Data from questionnaires administered when the index child was 4y or 6y were used to derive the exposure variables, ages of children and number of children (see Table 1).

**Covariates and descriptor variables.** To identify covariates to include in models as potential confounders and competing exposures, direct acyclic graphs (DAGs) were created for both the association between ages of children and maternal MVPA and LMVPA (Ages of Children Analyses) and number of children and maternal MVPA and LMVPA (Number of Children Analyses) [23] (see S2 and S3 Appendices). Variables assessed in the DAGs were living with the father of the index child, maternal education, maternal age, maternal body mass index (BMI), maternal employment and pre-school attendance.

Data collected at age 4y or 6y and earlier waves were used to derive covariates. Unless otherwise stated, they were derived from data at time of accelerometer data collection. Potential confounders were maternal age (continuous), maternal highest qualification level from

**Table 1. Derivation of exposure variables (ages of children and number of children categories).**

| | **How exposure variable was derived** | **Categories created** |
|---|---|---|
| **Ages of children** | Mothers reported how many younger/older children than the index child were in the household at the age 4 and age 6 surveys. | Younger children only; school-aged children only; children in both age groups. |
| | *For the age 4y survey*, all children younger than the index child were deemed $\leq$4y (younger children) and all children older than the index child were deemed $\geq$5y (school- aged). Mothers were then categorised as to whether they had only younger children or both school-aged and younger children (including the index child). | |
| | *For the age 6y survey*, all children older than the index child and the index child were allocated to the school-aged category. A household grid from the survey administered at age 2y (containing ages of all children in the household at that time) was used to assess age of the younger sibling when the index child was 6y. If a younger sibling was not mentioned at age 2y, and so their exact age could not be deduced, it was assumed that this sibling was $\leq$4y (a younger child) at the age 6 survey. Mothers were then categorised as to whether all their children were school-aged or whether they had both school-aged and younger children. | |
| **Number of children** | Mothers reported how many younger/older children than the index child were in the household at the age 4y or age 6y survey. At each of the two timepoints respectively (age 4y or age 6y), number of children was summed, including the index child. | 1 child; 2 children; $\geq$3 children. ($\geq$2 children in this paper refers to 2 or $\geq$3 children) |

pre-pregnancy data (none; certificate of secondary education; O-levels, A-levels. higher national diploma; degree), and living with the father of the index child (lives with father; does not live with father).

For all analyses, competing exposures included in models were season when accelerometry was conducted (winter; spring; summer; autumn), time of the week (weekday; weekend), and survey (age 4y; age 6y).

Additional descriptor variables were: self-reported employment (unemployed; employed), and maternal BMI and BMI categories according to World Health Organisation classifications [24] (underweight/ normal weight; overweight; obese) from measured height and weight data. Both were collected at the relevant age (4y or 6y).

## Statistical analyses

Analyses were conducted using Stata 16 [25]. Descriptive characteristics were calculated by ages of children. Means and SD were presented for normally distributed continuous variables and median and interquartile range (IQR) otherwise. To assess the representativeness of this sample, women in these analyses were compared with those who did not have any accelerometer data with respect to the following variables: maternal age at birth of index child, BMI at initial survey, and highest qualification level from the initial survey.

Two-level random intercept linear models were run, using daily minutes of maternal MVPA (log-transformed) and LMVPA as outcome variables. Minutes of MVPA were log-transformed as the distribution was skewed. Models were used to assess the association between ages of children or number of children and daily maternal log MVPA and LMVPA. Two-level random intercept linear models were used as they take into account variation in the outcome variable between day of accelerometer data collection (level 1) and between mothers (level 2) [26]. The correlation between observations was taken into account by allowing the intercept to vary randomly between mothers [13]. The model also takes into account that data from the age 4y and age 6y surveys were pooled and some mothers appear in both datasets.

An interaction term was added between number and ages of children using a likelihood ratio test (LRT) to compare models both with and without this term (Interaction Analyses). Analyses were stratified where an interaction was identified (defined a priori as p<0.05).

For MVPA, regression coefficients were back-transformed to produce the geometric mean ratio (GMR) for which a deviation from 1 indicates the percentage change in MVPA compared with the reference category for each analysis. Percentage difference (and 95% confidence interval (95%CI)) compared with the reference category is presented for each of the MVPA outcomes. For LMVPA, beta coefficients for mean daily minutes and 95%CI are presented for each of the outcomes.

Models were adjusted for competing exposures detailed earlier and confounders identified from the DAGs. Confounders were maternal age and number of children (Ages of Children Analyses), and maternal age, maternal education level and living with the father of the child (Number of Children Analyses and Interaction Analyses). A complete case analysis was conducted for each of the three individual analyses. Fig 1 shows the flow chart for mothers included in analyses. 848 women were included in analyses related to the ages of children and 844 in analyses related to number of children in the household.

For the periods of the day exploratory analyses, data were restricted to those women with 17 hours per day of accelerometer data in order to facilitate comparisons between parts of the day.

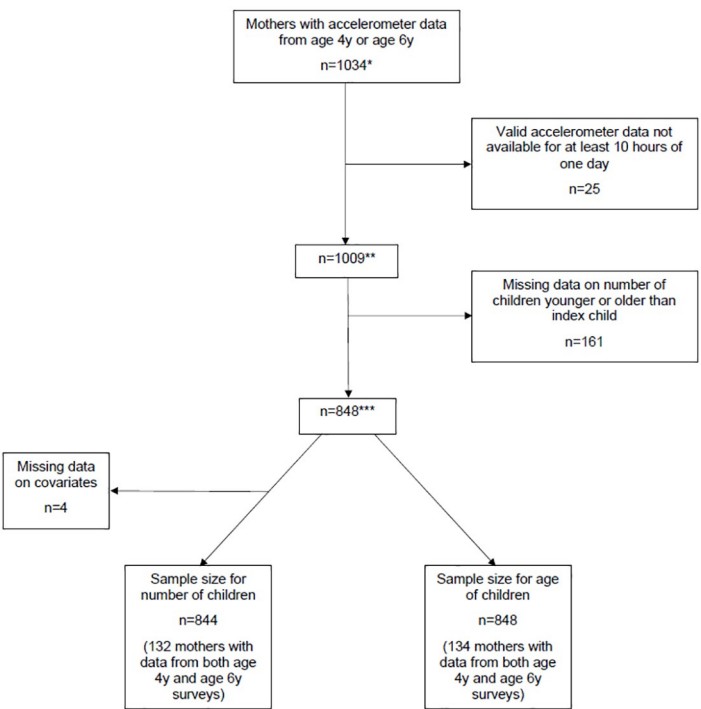

**Fig 1. A flow diagram chart of SWS mothers in analyses.** *n = 621 for age 4y and n = 608 for age 6y. **n = 607 for age 4y and n = 592 for age 6y. ***n = 457 for age 4y and n = 525 for age 6y.

## Sensitivity analyses

To assess the impact of the accelerometer inclusion criteria, Number of Children Analyses and Ages of Children Analyses for all days available were re-run, first for women with 3 or more valid days of accelerometer data (n = 801 for Ages of Children Analyses and n = 797 for Number of Children Analyses), and then for women with 5 or more valid days (n = 725 for Ages of Children Analyses and n = 721 for Number of Children Analyses). We excluded women with missing data for number of younger or older children, conducting sensitivity analyses with the assumption that those women had no children in these groups. The statistical analysis plan also included a complete case analysis for women with sufficient data for all three sets of analyses (Ages of Children, Number of Children and Interaction Analyses), but as only 4 women would have been excluded, these were not conducted.

## Results

Tables 2 and 3 show the descriptive characteristics of mothers by ages of their children for those mothers contributing data from the age 4y and age 6y surveys respectively. In all groups, less than 50% of mothers met the physical activity guidelines (29.8% of mothers with only younger children and 41.6% with younger and school-aged children at the age 4y survey (Table 2); 49.3% with only school-aged children and 39.8% with younger and school-aged children at the age 6y survey (Table 3). Median valid wear-time of the days included was 16.3 hours (IQR 16.0–17.0 hours) and median valid days was 6 (IQR 6–7) at the age 4y survey. The respective values at the age 6y survey were 16.3 (16.0–16.9) hours and 6 (IQR 5–7) days.

S4 Appendix shows a comparison of descriptive characteristics between women included in analyses (n = 848) and those excluded (n = 2124). Mothers were comparable in age at birth of

**Table 2. Descriptive characteristics of SWS mothers providing data at the age 4y survey (n = 457)[*].**

| | Ages of children | |
| --- | --- | --- |
| | Younger children only (n = 238) | Younger and School-aged children (n = 219) |
| Age, years (mean, SD) | 34.1 (3.4) | 35.4 (3.6) |
| BMI, kg/m$^2$ (median, IQR) | 25.0 (22.3–28.8) | 25.3 (22.4–29.1) |
| BMI category (%, n) | | |
| Normal weight or underweight | 50.0 (119) | 46.1 (101) |
| Overweight | 28.2 (67) | 34.7 (76) |
| Obese | 21.9 (52) | 19.2 (42) |
| Highest qualification level (%, n) | | |
| A levels or above | 68.1 (160) | 57.1 (125) |
| Living with father (%, n) | 93.3 (221) | 87.2 (191) |
| Unemployed (%, n) | 24.9 (59) | 30.1 (65) |
| Number of children (%, n) | | |
| 1 | 34.9 (83) | n/a |
| 2 | 59.7 (142) | 57.5 (126) |
| ≥3 | 5.5 (13) | 42.5 (93) |
| **Physical activity characteristics** | | |
| LMVPA (mins/day) (mean, SD) | | |
| All days | 432.8 (96.1) | 426.7 (97.6) |
| Weekdays | 433.5 (100.5) | 434.4 (101.5) |
| Weekend days | 432.6 (105.0) | 408.6 (113.7) |
| MVPA (mins/day) (median and IQR) | | |
| All days | 14.7 (8.5–22.7) | 17.7 (10.3–27.6) |
| Weekdays | 15.0 (8.6–25.8) | 18.5 (12.0–31.6) |
| Weekend days | 9.0 (3.0–17.8) | 10.6 (3.5–21.5) |
| Meeting physical activity guidelines (%, n) | 29.8 (71) | 41.6 (91) |

[*]The sample size here refers to that for ages of children and maternal physical activity analyses.

BMI = Body Mass Index; IQR = interquartile range; LMVPA = light, moderate or vigorous physical activity; MVPA = moderate or vigorous physical activity; SD = standard deviation. Meeting the physical activity guidelines is defined as ≥21.4 mins per day (≥150 mins/week) MVPA. Missing values: age 0%, BMI 0%, highest qualification level <1% (n<10), lives with father <1% (n<10), employment <1% (n<10).

the index child and BMI, but those included were more likely to have achieved a higher level of educational qualification than those excluded.

No interaction was found between ages of children and number of children in relation to maternal MVPA (as evidenced by a LRT p-value of 0.32). However, a strong interaction was found for LMVPA (as evidenced by a LRT with p = 0.0006). Thus, all results for MVPA are unstratified and those for LMVPA are stratified (ages of children by number of children and vice-versa).

## Association between ages and number of children and maternal MVPA (Table 4)

Mothers either with only school-aged children or children in both age groups did more MVPA than those with only younger children (by 46.9% [22.0, 77.0] and 42.7% [25.1, 62.8] respectively). There was stronger evidence for this association on weekdays than weekend

**Table 3. Descriptive characteristics of SWS mothers providing data at the age 6y survey (n = 525)[*].**

| | School-aged children only (n = 304) | Younger and school-aged children (n = 221) |
|---|---|---|
| Age, years (mean, SD) | 37.5 (3.9) | 36.3 (3.6) |
| BMI, kg/m$^2$ (median, IQR) | 25.4 (23.1–29.2) | 25.2 (22.3–28.4) |
| BMI category (%, n) | | |
| Normal weight or underweight | 43.4 (132) | 48.9 (108) |
| Overweight | 36.2 (110) | 33.5 (74) |
| Obese | 20.4 (62) | 17.7 (39) |
| Highest qualification level (%, n) A levels or above | 55.1 (167) | 67.7 (149) |
| Living with father (%, n) | 79.9 (243) | 93.7 (207) |
| Unemployed (%, n) | 16.2 (49) | 31.5 (69) |
| Number of children (%, n) | | |
| 1 | 23.7 (72) | n/a |
| 2 | 57.6 (175) | 62.4 (138) |
| ≥3 | 18.8 (57) | 37.6 (83) |
| Physical activity characteristics | | |
| LMVPA (mins/day) (mean, SD) | | |
| All days | 404.6 (96.2) | 426.3 (100.6) |
| Weekdays | 414.4 (98.7) | 435.3 (105.4) |
| Weekend days | 390.4 (113.3) | 404.3 (116.6) |
| MVPA (mins/day) (median and IQR) | | |
| All days | 21.1 (12.8–35.8) | 18.5 (10.6–28.5) |
| Weekdays | 23.8 (13.0–41.4) | 20.8 (12.1–33.1) |
| Weekend days | 11.8 (4.5–25.5) | 9.0 (3.5–19.0) |
| Meeting physical activity guidelines (%, n) | 49.3 (150) | 39.8 (88) |

[*]The sample size here refers to that for ages of children and maternal physical activity analyses.

BMI = Body Mass Index; IQR = interquartile range; LMVPA = light, moderate or vigorous physical activity; MVPA = moderate or vigorous physical activity; SD = standard deviation. Meeting the physical activity guidelines is defined as ≥21.4 mins per day (≥150 mins/week) MVPA. Missing values: age 0%, BMI 0%, highest qualification level <1% (n<10), lives with father 0%, employment <1% (n<10).

days. Regarding number of children, compared with mothers who had only 1 child, mothers with ≥2 children did less MVPA (by 12.5%, [-1.1, 24.3] for those with 2 children, and by 13.9% [-1.4, 26.9] for ≥3 children). The evidence for less MVPA amongst women with multiple children was stronger on weekend days than weekdays.

## Association between ages of children and maternal LMVPA by number of children (Fig 2)

For mothers with only one child, there was no evidence of a difference in LMVPA by age of the child. For mothers with 2 children, those with only school-aged children did 34.0 [3.9, 64.1] minutes less LMVPA than those with only younger children. For mothers with ≥3 children, those with any school-aged children did less LMVPA than those with only younger children. There was stronger evidence for these associations on weekend days than weekdays (see S5 Appendix).

**Table 4. Associations between number and ages of children and maternal MVPA levels[a].**

| | Percentage difference [95%CI] in MVPA[b] | | | | | |
|---|---|---|---|---|---|---|
| | **All days** | | **All weekdays** | | **All weekend days** | |
| **Ages of children (ref: younger children only)** | | | | | | |
| School-aged only | 46.9 [22.0, 77.0] | p<0.01 | 63.1 [34.0, 98.5] | p<0.01 | 39.2 [4.8, 84.9] | p = 0.02 |
| Both age groups | 42.7 [25.1, 62.8] | p<0.01 | 55.7 [35.5, 78.9] | p<0.01 | 25.2 [1.2, 54.9] | p = 0.04 |
| **Number of children (ref: 1 child)** | | | | | | |
| 2 children | -12.5 [-24.3, 1.1] | p = 0.07 | -10.0 [-22.7, 4.8] | p = 0.18 | -14.9 [-30.7, 4.4] | p = 0.12 |
| ≥3 children | -13.9 [-26.9, 1.4] | p = 0.07 | -10.5 [-24.6, 6.3] | p = 0.21 | -21.9 [-37.9, -1.8] | p = 0.03 |

[a]845 mothers were included in weekday and 768 in weekend analyses related to the ages of children. 841 mothers were included in weekday and 764 in weekend analyses related to the number of children.

[b]Percentage difference in MVPA is calculated from the geometric mean ratio as MVPA was log-transformed for analyses.

Ages of children models adjusted for age of mother, number of children, season, age 4y or age 6y survey, time of week (for all days analysis). Number of children models adjusted for age of mother, maternal highest qualification level, living with father, season, age 4y or age 6y survey, time of week (for all days analysis). 95%CI = 95% confidence interval; MVPA = moderate or vigorous physical activity.

## Association between number of children and maternal LMVPA by ages of children (Fig 3)

For mothers with only younger children, those with ≥2 children engaged in more minutes of LMVPA per day compared with those with one child (on average 42.6 extra minutes for mothers with 2 children; 49.9 extra minutes for those with ≥3 children), with the effect being more prominent on weekdays (S5 Appendix). For mothers with only school-aged children, there was no evidence of a difference in maternal LMVPA by number of children. For mothers with children in both age groups, having ≥3 children compared with 2 children was associated with more maternal LMVPA, at least on weekdays (see S5 Appendix).

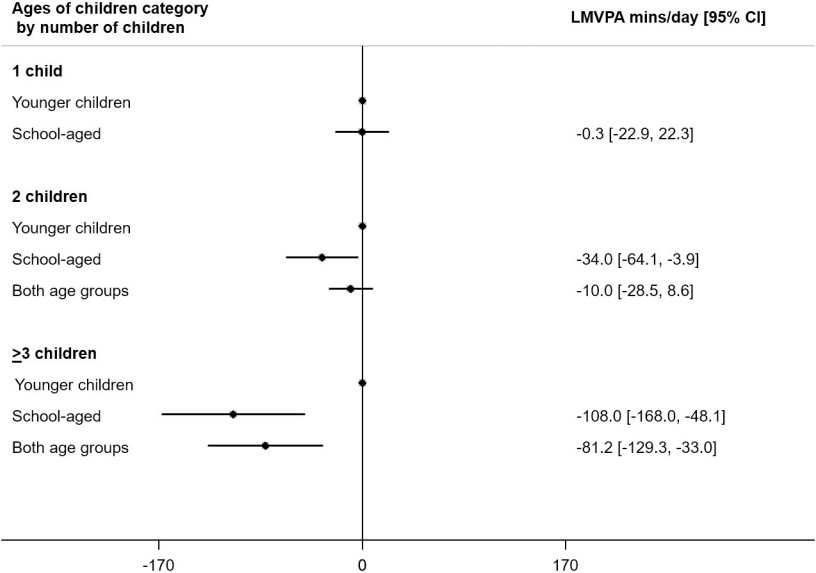

**Fig 2. Association between ages of children and maternal LMVPA by number of children.** LMVPA = light, moderate or vigorous physical activity; 95%CI = 95% confidence interval. Models adjusted for age of mother, season, age 4y or age 6y survey, time of week.

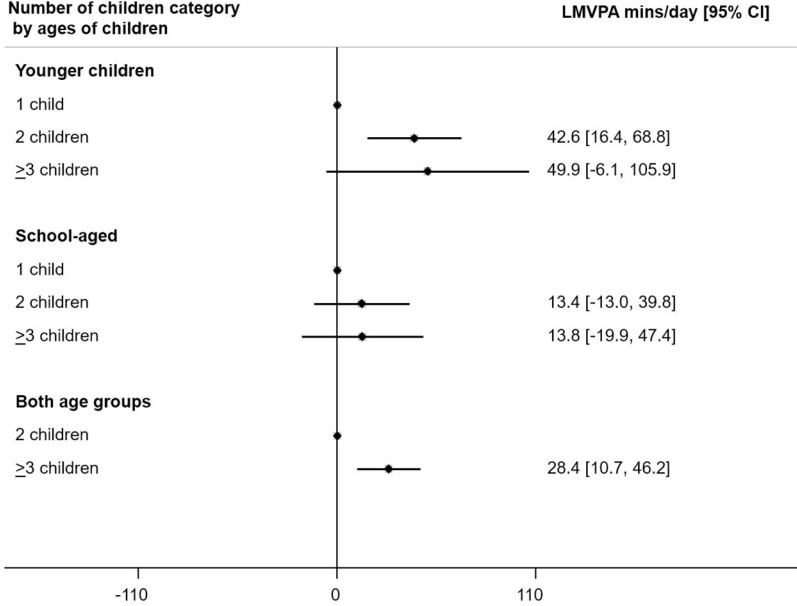

**Fig 3. Association between number of children and maternal LMVPA by ages of children.** LMVPA = light, moderate or vigorous physical activity; 95%CI = 95% confidence interval. Models adjusted for age of mother, maternal highest qualification level, living with father, season, age 4y or age 6y survey, time of week.

## Exploratory analyses for maternal MVPA by time of the day (S6 Appendix)

Mothers with any school-aged children did more MVPA than those with only younger children across all periods of the day and week, with the most pronounced differences on weekday mornings and weekday late afternoons. Regarding number of children, mothers with ≥2 children did less MVPA than mothers with 1 child, particularly on weekday mornings and weekend days.

## Exploratory analyses for maternal LMVPA by time of the day (S7 Appendix)

The associations between ages and number of children and maternal LMVPA varied by period of the day on weekdays and weekend. However, there were no clear patterns to support consistent conclusions.

## Sensitivity analyses

The pre-planned sensitivity analyses did not substantially affect the main findings (S8 and S9 Appendices).

## Discussion

### Main findings

Amongst this cohort of British women, we found that mothers with any school-aged children engaged in more MVPA than those with only younger children, and that mothers with multiple children did less MVPA than those with one child. In addition, for mothers with multiple children, those with any school-aged children did less LMVPA than those with only younger

children. Finally, among mothers with any younger children, those with more children did more LMVPA.

## Findings in relation to other similar studies

Only one other study has used accelerometer assessment to directly examine the association between ages of children and individual physical activity amongst mothers, and these analyses were only exploratory [10]. Although significant differences in MVPA were not reported between mothers of children of different ages, and the youngest age category was 0–5 years rather than 0–4 years in our study, their estimates did suggest that mothers of children aged 0–5 years did less MVPA (22.9 mins/day) than those with children aged 6–12 years (28.7 mins/day) [10], supporting our findings. However, unlike in our study in which the average amount of MVPA was below the recommended guidelines regardless of child age (see Tables 2 and 3), the average amount of MVPA for all mothers in Candelaria et al. 2012 was above the recommended guidelines [10]. Although they did not make direct comparisons between groups of mothers, two other studies using accelerometer assessment compared the physical activity levels of mothers with children in different age groups to those of non-mothers and showed that only mothers with children ≤5y did significantly less MVPA than non-parents [11, 12]. Again, despite the youngest category of children extending to 5-year-olds rather than 4-year-olds as in our study, this also supports our finding of mothers with younger children doing less MVPA than those with older children. In the one study that investigated accelerometer-assessed LPA by ages of children [12], women with children ≤5y were also the only group of mothers who had significantly higher levels of LPA compared with non-parents, which aligns with our findings of mothers of younger children doing more LMVPA than those with school-aged children. To our knowledge, there have been no recent relevant self-report studies comparing maternal MVPA or LMVPA by whether children in the household are younger children or school-aged.

Regarding number of children, only one study used accelerometer assessment to directly investigate a statistical association with maternal physical activity amongst a sample of parents, finding no association [10]. Two studies using accelerometer assessment also made direct comparisons between mothers with different numbers of children and non-parents; one study found no association [11], while the other, in support of our findings, reported that mothers with ≥2 children did less MVPA than non-parents, whilst there was no difference for those with one child [12]. In the only study to examine accelerometer assessed LPA [12], estimates suggested that there was little or no difference in LPA by number of children, in contrast to our findings, but again direct comparisons were only made between mothers and non-parents. The majority of recent studies using self-reported physical activity have found no difference in maternal physical activity, but outcomes have varied: total physical activity [27], active transport [28] or leisure time physical activity [29]. One study did find that parents with multiple children do less exercise than those with one child, which supports our findings [30].

## Possible explanations for our findings

Mothers with younger children spent less time in MVPA but more in LMVPA than those with school-aged children. Evidence suggests that mothers of young children spend a large amount of time in household activities [31], which tend to be of lower intensity. As children grow older, mothers may spend less time purely supervising their children as they play [32], instead engaging in higher intensity activity through co-participation with their children and actively traveling to and from school. These opportunities for increased MVPA would be available to those with purely school-aged or both younger and school-aged children. Having children at

school may also provide mothers with more time to engage in structured exercise of higher intensity. Outside school, mothers may feel more comfortable leaving older children with someone else to do their own leisure time physical activity. As children get older, women are also more likely to return to work, providing opportunities for higher intensity physical activity both during the commute, and at work itself in some cases.

Less time spent in MVPA amongst mothers with multiple children could be due to greater time constraints, reducing time available for leisure time physical activity, and perhaps encouraging mothers to avoid active transport in favour of quicker options. More time spent in LMVPA amongst mothers with more children could be explained by additional time spent on household activities, supported by a study which found that mothers of one child spend on average 268 minutes per week on household activities compared with 520 minutes per week amongst mothers with ≥3 children [10].

## Strengths and limitations

This is the largest study to assess the association between number and ages of children and accelerometer-assessed maternal physical activity, benefits from the inclusion of testing for statistical interactions, and makes comparisons between groups of mothers as opposed to comparing with non-parents. In addition, the use of accelerometer assessment of physical activity is a key strength, important to ensure that sporadic and spontaneous movement was captured, which is likely to be common amongst mothers of younger children and might be easily missed in self-reported physical activity [33]. The physical activity data is from 2006 to 2012 but is still highly relevant considering that physical activity levels amongst adults in high-income western countries have since continued to decline [34]. Finally, our analyses were detailed, not only accounting for the interaction between number and ages of children in relation to maternal LMVPA and including appropriate confounders, but also exploring associations by period of the day and week.

We lacked data on exact ages of the siblings, and assumed that younger siblings of the 6y index child were ≤4y (i.e. in the younger children category) if they were not mentioned at the age 2y survey. As the average time lapse between the age 2y and age 6y survey was 4.6y, we think this was a reasonable assumption. We lacked data on sex of children other than the index child so this variable was not included in the models. We also do not know the types of physical activity the women engaged in or their locations whilst wearing the accelerometer, and we were unable to consider all factors which might be related to maternal physical activity, such as size of garden, and social support. Finally, due to the number of tests conducted, some of our findings, especially those from exploratory analyses, may be due to chance.

## Policy implications and future research needed

Our findings suggest a need to focus on interventions and policies to increase the opportunities for higher intensity physical activity of mothers of younger or multiple children, as these mothers already undertake considerable amounts of LMVPA. LPA does provide health benefits, but in a study examining the association between device-assessed physical activity and mortality rates, greater reductions were achieved amongst those reaching a set physical activity energy expenditure through higher intensity physical activity than through more time spent in lower intensity activity [19]. Qualitative research may be helpful to identify means by which this change could be achieved. Large longitudinal studies using device-based assessment of physical activity are also required to build the evidence base relating to determinants of change in maternal physical activity, tracking changes in individuals as their children get older. Studies assessing physical activity through a combination of device-assessment and self-report would

also be helpful to provide information on both amount and type of physical activity engaged in by mothers. Finally, research is needed to investigate the association between number and ages of children and paternal physical activity, to identify fathers more at risk of insufficient physical activity and to determine how this relates to both maternal and child activity.

## Conclusions

Policies and interventions are needed to encourage mothers with younger or multiple children to engage in greater MVPA, ensuring they benefit from health gains associated with higher intensity activity.

## Supporting information

**S1 Appendix. STROBE statement.**
(DOCX)

**S2 Appendix. DAG for number of children and maternal physical activity analyses.**
(PDF)

**S3 Appendix. DAG for ages of children and maternal physical activity analyses.**
(PDF)

**S4 Appendix. A comparison of descriptive characteristics.**
(DOCX)

**S5 Appendix. Stratified associations between exposures and maternal LMVPA.**
(DOCX)

**S6 Appendix. Associations between exposures and MVPA by time of the day.**
(DOCX)

**S7 Appendix. Stratified associations between exposures and LMVPA by time of the day.**
(DOCX)

**S8 Appendix. Sensitivity analyses for MVPA.**
(DOCX)

**S9 Appendix. Sensitivity analyses for LMVPA.**
(DOCX)

## Acknowledgments

We thank the participants in the SWS for their commitment to and involvement in the study, and the dedicated team of research nurses and ancillary staff for their assistance in collecting and processing the data. The authors would also like to thank Stephanie Hollidge (MRC Epidemiology Unit) for her assistance in the processing of the physical activity data.

## Author Contributions

**Conceptualization:** Rachel F. Simpson, Kathryn R. Hesketh, Esther M. F. van Sluijs.

**Data curation:** Sarah R. Crozier, Janis Baird, Cyrus Cooper, Keith M. Godfrey, Nicholas C. Harvey, Kate Westgate, Hazel M. Inskip.

**Formal analysis:** Rachel F. Simpson, Kathryn R. Hesketh, Kate Westgate.

**Funding acquisition:** Sarah R. Crozier, Janis Baird, Cyrus Cooper, Keith M. Godfrey, Nicholas C. Harvey, Hazel M. Inskip.

**Investigation:** Sarah R. Crozier, Janis Baird, Cyrus Cooper, Keith M. Godfrey, Nicholas C. Harvey, Hazel M. Inskip.

**Methodology:** Rachel F. Simpson.

**Project administration:** Sarah R. Crozier, Janis Baird, Cyrus Cooper, Keith M. Godfrey, Nicholas C. Harvey, Hazel M. Inskip.

**Supervision:** Kathryn R. Hesketh, Esther M. F. van Sluijs.

**Writing – original draft:** Rachel F. Simpson.

**Writing – review & editing:** Rachel F. Simpson, Kathryn R. Hesketh, Sarah R. Crozier, Janis Baird, Cyrus Cooper, Keith M. Godfrey, Nicholas C. Harvey, Kate Westgate, Hazel M. Inskip, Esther M. F. van Sluijs.

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
