## [Decision Letter · Decision Letter 0]

13 May 2022

PONE-D-22-10284The association between number and ages of children and the physical activity of mothers: cross-sectional analyses from the Southampton Women’s SurveyPLOS ONE

Dear Dr. Simpson,

Thank you for submitting your manuscript to PLOS ONE. After careful consideration, we feel that it has merit but does not fully meet PLOS ONE’s publication criteria as it currently stands. Therefore, we invite you to submit a revised version of the manuscript that addresses the points raised during the review process.

We look forward to receiving your revised manuscript.

Kind regards,

Frank T. Spradley

Academic Editor

PLOS ONE

Journal Requirements:

[I have read the journal's policy and the authors of this manuscript have the following competing interests: CC received personal fees from Alliance for Better Bone Health, Amgen, Eli Lilly, GSK, Medtronic, Novartis, Pfizer, Roche, Servier, Takeda and UCB. KMG has received reimbursement for speaking at conferences sponsored by companies selling nutritional products, and is part of an academic consortium that has received research funding from Abbott Nutrition, Nestec, BenevolentAI Bio Ltd. and Danone. NCH has received consultancy, lecture fees and honoraria from Alliance for Better Bone Health, AMGEN, MSD, Eli Lilly, Servier, UCS, Shire, Consilient Healthcare, Kyowa Kirin and Internis Pharma. The remaining authors declare they have no competing interests.] 

Reviewers' comments:

Reviewer's Responses to Questions

**Comments to the Author**

1. Is the manuscript technically sound, and do the data support the conclusions?

Reviewer #1: Yes

Reviewer #2: Yes

Reviewer #3: Yes

Reviewer #4: Yes

2. Has the statistical analysis been performed appropriately and rigorously? 

Reviewer #1: Yes

Reviewer #2: Yes

Reviewer #3: Yes

Reviewer #4: I Don't Know

3. Have the authors made all data underlying the findings in their manuscript fully available?

Reviewer #1: No

Reviewer #2: Yes

Reviewer #3: Yes

Reviewer #4: Yes

4. Is the manuscript presented in an intelligible fashion and written in standard English?

Reviewer #1: Yes

Reviewer #2: Yes

Reviewer #3: Yes

Reviewer #4: Yes

5. Review Comments to the Author

Reviewer #1: The association between number and ages of children and the physical activity of mothers: cross-sectional analyses from the Southampton Women’s Survey

This manuscript investigates the physical activity behaviours of mothers, stratified by the number and ages of children. Accelerometer-assessed minutes of physical activity were analysed at two timepoints (when the index child was 4 years (n=457) and 6 years (n=525)) and enabled further comparison of physical activity patterns between mothers with school-aged (≥5 years) and younger (≤4 years) children. Mothers with multiple children and those with any children aged ≤4 years did less MVPA, across all periods of the day and week. When mothers had multiple children with any school-aged, there was a lower performance of LMVPA. However, when mothers had multiple only younger children, there was an increase in LMVPA.

Broad comments

This manuscript proposes the need to focus on interventions and policies to increase the opportunities for higher intensity physical activity for mothers with multiple or younger children. Indeed, this manuscript provides evidence to show that specific cohorts of mothers fail to engage in sufficient physical activity and can be related to the number and age/s of their children. However, further discussion of the potential health impacts of lower volume and/or lower intensity physical activity in this population is required to highlight the significance of this research. The statistical analysis is sound; however, the authors could consider the use of an R-squared value to identify how much variation in the outcome measures are explained by physical activity and the other covariates.

Specific comments

Line 70 – Perhaps be more specific on what type (i.e. intensity) of physical activity is associated with these benefits as the discussion separates LMVPA from MVPA, with conclusions emphasising the need for performance of higher intensity activity.

Line 98-99 – Please describe either here or in the methods (line 151-152) the benefits of LPA vs MVPA to 1) highlight the difference between the two intensities and 2) set the background to support the conclusion that higher intensity activity opportunities are needed for mothers. It has been identified that LPA has health benefits (line 158), but it should be explicitly discussed if the benefits are inferior or superior to MVPA or whether the volume of physical activity needs to be considered when performing LPA.

Line 156 – Why was LMVPA used as an outcome as opposed to just LPA? Does the addition of MVPA in the measure add any value when it is LPA that is the main concern of this measure? Could there be an interaction effect between LPA and MVPA?

Line 162 – Reference #20 specifies that 100 counts per minute is used to determine sedentary time but has been used as a threshold for LPA in the current study. Sedentary time (≤1.5 METs) is typically separate to light/low intensity physical activity (1.5-3 METs). Indeed, LMVPA is ≥100 counts, so perhaps needs further clarification of this.

Line 173 – Covariates – why is BMI not included as a confounder in the model as BMI categories can influence physical activity levels? Has maternal occupation been considered? Full-time vs part-time vs unemployed, office job vs on their feet.

Line 252 – Please specify sample sizes for age 4y and 6y in each level of the flow diagram chart.

Line 299 – For consistency, consider use of just tables or graphs for presenting associations between number and ages of children and MVPA or LMVPA (i.e. table 4, figure 2 and 3).

Line 373 – The group that performed less MVPA still met the physical activity guidelines, is this comparable to the current study? It is worth addressing whether the groups stratified by age/number of children, when describing the percentage change in physical activity, met the physical activity guidelines.

Line 399 – Consider highlighting the limitations of self-report data vs accelerometer assessed data.

Line 425 – One potential limitation to this study is the representativeness of this study cohort to the current population of mothers, given the changes in physical activity culture and opportunities in 2022 vs 2000-2012. Please address this.

Line 448-449 – Please discuss in further detail why women who already undertake large amounts of LPA (likely involving playing with and picking up young children) need to engage in additional MVPA? Is this for health benefits that are only related to MVPA rather than LPA? Is LPA of mothers different to LPA of non-parents in terms of intensity and duration? Could the volume of LPA be enough to see other health benefits while also being more achievable and feasible for mothers?

Reviewer #2: This is a well-conducted observational study investigating the association between number and ages of children and the physical activity of mothers based on cross sectional analyses. The authors were interested in investigating this due to previous small number of works, which were only exploratory and showed significant differences in MVPA but were not reported between mothers of children of different ages. This study is conducted appropriately with good sound research design and appropriate reason to conduct the study.

Howerver I have some minor comments:

Covariates and descriptor variables

The list of variables was established on the basis of a path analysis. Please complete the analysis with the ‘p’ values or the values of β coefficients for the relationships between covariance and dependent and independent variables. I think that for a reader with an average level of statistical knowledge, such a figure does not clearly explain how covariates were selected (description of relationships under the figure - would be a good idea).

What was the reason for the ommition of the sex of children in statistical analysis?

Statistics

220 – what was the reference group? Enter in parentheses.

Results

248-251. Fig 1. Flow chart does not apply to results. Move it to the material and methods section.

Tables

Tables do not present the exact data, please put the p values. Please take into account that most often the reader who wants to learn about the analyzes usually uses tables and figures. Hence tables must show the data clearly and precisely.

Disscusion

My comment concerns ‘possible explanations for our findings’. It was emphasized that the only significant explanation for the low physical activity of mothers of younger children and mothers of multiple children were their long, low-activity housework and childcare activities. Was it possible that other factors have played a role in maternal PA (for example: family’s income, motivation to excercise, mother’s level of education, and awareness of the importance of PA, support of other family members, BMI, and wide range of variables related to children’s: sex, sibling relations in the context of parents PA, specificity of the development of young children, their level of motor abilities…. ). Is it possible to elaborate on that?.

Appedix

I have included comments about the path analysis and tables earlier.

Reviewer #3: Thank you for giving me the opportunity to review this manuscript. The approach with the use of an accelerometer in the assessment of PA is rare so the material becomes more interesting. But it needs more clarification whether this double publication or not because part of this result like ''Women with any school-aged children engaged in more MVPA than those with only <4y e.g.% difference in minutes of MVPA [95%CI]: 42.2% [18.7-70.4] for mothers with only school aged vs only <4y). Mothers with multiple children did less MVPA than those with 1 child (e.g.51 13.9% [1.0-25.2] less MVPA for those with 2 children almost similar to the result published by the same authors titled as''Cross-sectional associations between number and ages of children and maternal physical activity ''Published: 20 October 2021 on European Journal of Public Health, Volume 31, Issue Supplement_3, October 2021, ckab164.789, https://doi.org/10.1093/eurpub/ckab164.789.

-Abstract

The abstract contains some short forms of words that were not written in the full form first.

-Introduction: In the first paragraph ref.1-3 all discuss the benefits of PA but ref 2 a number of times. It is better if ideas merged together and are referenced at last.

-I suggest supporting your conclusion part in the main document at the end with your finding than mentioning your title

Reviewer #4: The authors have taken a very important and novel aim to investigate the associations between ages and number of children and device-measured maternal physical activity (PA). This study also aimed at identifying the subgroups of mothers who are more at risk of insufficient PA than others. This work has a great practical value. The authors made a huge research effort to collect data over so many years.

However in my opinion some corrections of this manuscript should be done:

1. Please be consistent with specifying the groups of children. In the line 42 you have: “the index child was aged 4y or 6y” and in lines 108-110: the aim of this cross-sectional study is to investigate the association between number and ages of children, categorized as >5y (hereafter referred to as school aged) and ≤ 4 y…. Even though the authors explains the categories of mothers in the Table 1, it is very confusing for the reader to understand.

2. I think it should be stated already in the abstract that the surveys were performed in two time points.

3. You surveyed the women between March 2006 and June 2009 and between March 2007 and August 2012. Were there women in this group who participated in the survey twice?

4. Lines 182-183: Please explain “maternal highest qualification level from pre-pregnancy data (6 categories ranging from none to degree) – what are these 6 categories?

5. I totally agree with the statement (430 – 433) that a key strength of this work was the use of accelerometers to assess PA, important to ensure that sporadic and spontaneous movement was captured, which is likely to be common amongst mothers of younger children and might be easily missed in self-reported PA. It would valuable to translate obtained outcomes to practical recommendations on how to increase PA levels in mothers from specific subgroups, e.g how to increase the MVPA of mothers with younger children. For future studies it would be also valuable to collect data (based on questionnaires) which kind of PA the mothers were involved in, e.g. transportation, household activities, supervised sessions, etc. It could be helpful to develop proper strategies to increase the PA level, especially in mothers who are more at risk of insufficient PA than others.

6. Please be consistent throughout the entire text to use the term Physical activity or the “PA” abbreviation.

7. Minor comment: I suggest to change the reference for an updated version of the WHO guidelines (WHO, 2020).

WHO. (2020). WHO guidelines on physical activity and sedentary behaviour. Geneva: World Health Organization. In (Vol. 57). Geneva: World Health Organization.

6. PLOS authors have the option to publish the peer review history of their article (what does this mean?). If published, this will include your full peer review and any attached files.

Reviewer #1: No

Reviewer #2: No

Reviewer #3: No

Reviewer #4: **Yes: **Anna Szumilewicz

---

## [Author Response · Author response to Decision Letter 0]

16 Aug 2022

Please see attached file for responses to reviewers. Thank you very much.

---

## [Decision Letter · Decision Letter 1]

8 Sep 2022

PONE-D-22-10284R1The association between number and ages of children and the physical activity of mothers: cross-sectional analyses from the Southampton Women’s SurveyPLOS ONE

Dear Dr. Simpson,

Thank you for submitting your manuscript to PLOS ONE. After careful consideration, we feel that it has merit but does not fully meet PLOS ONE’s publication criteria as it currently stands. Therefore, we invite you to submit a revised version of the manuscript that addresses the points raised during the review process.

We look forward to receiving your revised manuscript.

Kind regards,

Frank T. Spradley

Academic Editor

PLOS ONE

Journal Requirements:

Reviewers' comments:

Reviewer's Responses to Questions

**Comments to the Author**

1. If the authors have adequately addressed your comments raised in a previous round of review and you feel that this manuscript is now acceptable for publication, you may indicate that here to bypass the “Comments to the Author” section, enter your conflict of interest statement in the “Confidential to Editor” section, and submit your "Accept" recommendation.

Reviewer #2: All comments have been addressed

Reviewer #3: All comments have been addressed

Reviewer #4: All comments have been addressed

2. Is the manuscript technically sound, and do the data support the conclusions?

Reviewer #2: Yes

Reviewer #3: Yes

Reviewer #4: Yes

3. Has the statistical analysis been performed appropriately and rigorously? 

Reviewer #2: No

Reviewer #3: Yes

Reviewer #4: Yes

4. Have the authors made all data underlying the findings in their manuscript fully available?

Reviewer #2: Yes

Reviewer #3: Yes

Reviewer #4: Yes

5. Is the manuscript presented in an intelligible fashion and written in standard English?

Reviewer #2: Yes

Reviewer #3: Yes

Reviewer #4: Yes

6. Review Comments to the Author

Reviewer #2: Thank you for replying to the review. I read them in detail. I agree with most. However, section 2.6 requires additional solutions. I agree that there are different approaches to statistical analyzes, nevertheless, the presented data should give us a full answer to research questions.

If the authors do not want to present the p-value, please also provide the efect size with a 95% CI. (or alternatively p-value). One of the two sollutions will be accepted.

I read the articles cited with interest, as well as others on p-value. The value of a 95% CI does not tell us about an estimate, just as the p-value does not give us an answer about the research hypothesis.

Reviewer #3: Thank you authors for your interesting work. All my comments have been adequately addressed. Now the paper can be acceptable for publication.

Reviewer #4: The authors have taken a very important and novel aim to investigate the associations between ages and number of children and device-measured maternal physical activity. This work has a great practical value and certainly should be published.

7. PLOS authors have the option to publish the peer review history of their article (what does this mean?). If published, this will include your full peer review and any attached files.

Reviewer #2: **Yes: **Elżbieta Cieśla

Reviewer #3: No

Reviewer #4: **Yes: **Anna Szumilewicz

---

## [Decision Letter · Decision Letter 2]

18 Oct 2022

The association between number and ages of children and the physical activity of mothers: cross-sectional analyses from the Southampton Women’s Survey

PONE-D-22-10284R2

Dear Dr. Simpson,

We’re pleased to inform you that your manuscript has been judged scientifically suitable for publication and will be formally accepted for publication once it meets all outstanding technical requirements.

Kind regards,

Subas Neupane

Guest Editor

PLOS ONE

Additional Editor Comments (optional):

Thank you for the revised manuscript which is much improved and I am pleases to accept it for publication in PLOS ONE.

Reviewers' comments:

Reviewer's Responses to Questions

**Comments to the Author**

1. If the authors have adequately addressed your comments raised in a previous round of review and you feel that this manuscript is now acceptable for publication, you may indicate that here to bypass the “Comments to the Author” section, enter your conflict of interest statement in the “Confidential to Editor” section, and submit your "Accept" recommendation.

Reviewer #2: All comments have been addressed

Reviewer #4: All comments have been addressed

2. Is the manuscript technically sound, and do the data support the conclusions?

Reviewer #2: Yes

Reviewer #4: Yes

3. Has the statistical analysis been performed appropriately and rigorously? 

Reviewer #2: Yes

Reviewer #4: Yes

4. Have the authors made all data underlying the findings in their manuscript fully available?

Reviewer #2: Yes

Reviewer #4: No

5. Is the manuscript presented in an intelligible fashion and written in standard English?

Reviewer #2: Yes

Reviewer #4: Yes

6. Review Comments to the Author

Reviewer #2: Dear Autors,

thank you for replying to my review. I read them in detail. I agree with all comments. I believe, your article meets all the criteria and should be appear in the scientific journal: PLOS ONE

Reviewer #4: Once again, I would like to congratulate the authors for their study. This work has a great practical value. The authors made a huge research effort to collect data over so many years.

7. PLOS authors have the option to publish the peer review history of their article (what does this mean?). If published, this will include your full peer review and any attached files.

Reviewer #2: No

Reviewer #4: **Yes: **Anna Szumilewicz

---

## [Editor Report · Acceptance letter]

20 Oct 2022

PONE-D-22-10284R2 

The association between number and ages of children and the physical activity of mothers: cross-sectional analyses from the Southampton Women’s Survey 

Dear Dr. Simpson:

I'm pleased to inform you that your manuscript has been deemed suitable for publication in PLOS ONE. Congratulations! Your manuscript is now with our production department. 

Kind regards, 

on behalf of

Dr. Subas Neupane 

Guest Editor

PLOS ONE